# Toward Integrated and Inclusive Education for Sustainability with School–University Cooperation

**Johanna Naukkarinen** [1,*] **and Laura Jouhkimo** [2]

1 School of Energy Systems, Lappeenranta-Lahti University of Technology LUT, 53851 Lappeenranta, Finland
2 Student Services, Lappeenranta-Lahti University of Technology LUT, 53851 Lappeenranta, Finland; laura.jouhkimo@lut.fi
* Correspondence: johanna.naukkarinen@lut.fi

**Abstract:** Sustainable development has been a topic in the Finnish core curriculum for several years, but integrating this cross-cutting theme into a subject-based curriculum is perceived to be difficult. Hence, the city of Lappeenranta has developed its own educational model to support the development of pupils' ecosocial attitudes and abilities. This study evaluates the implemented educational model by empirically examining the pupils' perceptions of sustainability as well as studying differences between different groups of pupils. Analysis methods consisted of a combination of factor analysis, linear regression, and statistical tests for group differences. Young people were discovered to hold three different kinds of orientations to environmental issues and ecological sustainability. These were coupled with different perceptions of science, technology, and business in a way that indicates three different types of perceptions of a more holistic conception of sustainability. The significant differences in the sustainability-related perceptions between girls and boys as well as between pupils with different educational aspirations suggest that in the future, special attention needs to be paid to develop the educational model for better equity and inclusivity.

**Keywords:** education for sustainable development; perceptions of sustainability; STEAM education; school–university cooperation; gender diversity; inclusion

## 1. Introduction

Sustainable development is a commonly used and broadly discussed concept in educational discourses. It has various meanings, definitions, and interpretations, not only in educational research but also in documents and principles guiding education providers, and at the level of everyday life in schools. The concept, used since the 1980s, arises from the growing awareness of the links between environmental problems, socioeconomic issues, such as poverty and inequality, and concerns about the future for humanity [1]. One of the most famous definitions of sustainable development was expressed in the Bruntland Report in 1987: sustainability means "meeting the needs of the present without compromising the ability of future generations to meet their own needs" [2].

Inclusive STEM education is believed to have plenty of potential for equitable education for sustainable education and social equity [3]. However, it requires change and innovation in the pedagogical approach and curriculum development [3]. The Uniori STEAM (science, technology, engineering, arts, mathematics) education model was built in the City of Lappeenranta in cooperation with the local university to provide accessible and inclusive STEAM education, which fosters the development of ecosocial attitudes and abilities. This paper examines the pupils' current perceptions of sustainability and evaluates the success and development needs of the Uniori model in the light of those perceptions. First, the concepts of sustainability education and the Uniori education model are described and discussed in more detail. Then, the empirical work on and results of sustainability perceptions are presented, followed by a discussion about their implications for the development of the Uniori model.

### 1.1. Education for Sustainable Development

A sustainable way of living is one of the key concepts in the Finnish national core curriculum for basic education, which has been in use since 2016 [4]. The necessity of a sustainable way of living has been considered one of the underlying values of the Finnish basic education. The basic education thus follows the principles of sustainable development and guides the pupils in adopting a sustainable way of living. All four dimensions of sustainable development—ecological, economic, social, and cultural—are acknowledged in the core curriculum [4]. The concept of a sustainable way of living is represented as ecosocial knowledge and ability; diversity and ability for renewal of ecosystems; building a competence base for circular economy; and a sustainable use of natural resources. Understanding the seriousness of climate change is also stressed in the core curriculum [4].

Sustainable development was one of the cross-cutting themes in the previous core curriculum for basic education, which was in use between 2004 and 2015 [5]. Although the cross-cutting themes were considered important by the teachers, 74% of them wanted to have them removed from the core curriculum as integrating the cross-cutting themes into the subject-based curriculum was perceived to be difficult [6]. In terms of subject objectives, the theme of sustainable development was most clearly connected with the curriculum in biology, geography, and civics [6]. A study of lower secondary school subject teachers' perceptions of their sustainable education competence and consideration of different dimensions of sustainability in their teaching revealed that biology, geography, and history teachers often used holistic sustainability approaches and included several dimensions of sustainability in teaching, whereas mathematics, physics, and chemistry teachers typically considered only the ecological dimension of sustainability [7]. Different orientations to the dimensions and holisticity of the sustainability approach can introduce tensions to cooperative planning and execution of sustainability education. Challenges arise when educators place a different emphasis between human developments and environmental conservation aspects, focus differently on the cognitive versus affective objectives of education, or disagree about the appropriate teaching methods [8].

One attempt of trying to cover the different orientations, dimensions, and approaches to sustainable development in education is the concept of ecosocial knowledge and ability, which has been part of the Finnish educational discourse for already more than a decade. The baseline of the concept is criticism of the materialistic and consumerist way of living, sustaining and developing democracy, and protecting the ecological foundations of life [9]. The core values of ecosocial knowledge and ability include a systemic worldview, responsibility, sufficiency, interpersonality, connectedness to nature, and future orientation [10]. Salonen and Bardy state that the hierarchy of perceiving the world according to the ecosocial knowledge and ability is (1) the priority of the ecological issues and securing the prerequisites of life for the future generations, (2) inalienability of human rights, and (3) fostering a stable economy to distribute the limited global resources as efficiently as possible and to secure the satisfaction of the basic needs for all [11]. Responsibility—also for future generations—and moderation are the core values for human growth and learning. Education must create hope, meaning, and value of life for children and young people [9]. At the age of alarming and wicked ecological, social, and economic problems, such as global warming, increasing inequality, and global pandemics, strengthening the participation of children and young people can also be considered important.

Much of the research of the sustainability-related values and attitudes of young people in Finland has concentrated on the ecological and environmental dimension of sustainability. At the beginning of the millennium, lower secondary school students were found to have positive attitudes toward environmental responsibility, which correlated strongly with naturocentric attitudes toward environmental values, or negative attitudes toward environmental responsibility, which correlated with anthropocentric attitudes toward environmental values [12]. The latter aspects were also linked to the reflection of environmental problems. In general, girls had higher positive attitude scores and lower

negative attitude scores than boys [12]. The study also discovered that positive attitudes toward environmental responsibility had a connection to general suspicion of science and technology and suggested that the relationship between these attitudes should be studied further [12].

Although there are several pedagogical options for holistic sustainability education through integration of different subjects [3], there are also considerable challenges, and teachers need support in both including the dimensions of sustainability in their own subject teaching and establishing constructed thematic teaching that integrates several subjects. To enhance education, a more comprehensive picture of pupils' attitudes and values toward different dimensions of sustainability and their interconnectedness is also needed. The next section describes a school–university cooperation model employed in Lappeenranta to support the teachers in science, technology, entrepreneurial, and sustainability education.

### 1.2. Uniori Model for Integrated and Inclusive School–University Cooperation

The Education Policy Report of the Finnish Government from 2021 states that as obvious a foundation for the new knowledge generated by scientific research is for higher education, scientific thinking can also serve as an actively integrated operating model in the entire education path. In the future, many citizens who do not have research training must obtain the skills needed for scientific research in order to reach the vision of "an exploring Finland of the next generation." According to the Report [13], "Systematic science education creates preconditions for a higher level of education and competence-based growth. The objective of science education is to advance and expand citizens' problem-solving ability and understanding of the operating methods, structures, and development of science."

One example of such operating models is Lappeenranta Junior University or Uniori, as it is locally called. Uniori is a joint STEAM education model that was designed by the City of Lappeenranta and LUT University in 2017. The objective of the cooperation is to increase children's and adolescents' interest in science, technology, research, and sustainable business, develop their skills for academic studies, and enhance their opportunities to build a sustainable future [14]. The activities also aim at engaging pupils' families to promote sustainability in their hometown [14]. All Uniori STEAM education activities are science-based and build on the strategic focus areas of LUT University: clean water, clean air, clean energy, and sustainable business. The scientific research of LUT university is seeking solutions for the reduction of climate change, promotion of wind and solar power, recycling of nutrients and waste materials, supply of clean water and energy, and sustainable renewal of businesses and society. These research areas are easy to combine with the values of the core curriculum of basic education and its striving for guiding the pupils in adopting sustainable living.

Every child is involved in Uniori STEAM education activities from early childhood education to high school. Annually, this means about 3000 pupils in preschools and basic education and about 1000 students in general upper-secondary education. All Uniori activities are included in the local curricula of the City of Lappeenranta, not only for basic education but also for early childhood and upper secondary education. Planning and implementation of the model are carried out in close cooperation with teachers, and all teachers whose classes are involved in the activities are educated every year. The concept is continuous and systematic, and instead of temporary project funding, it is funded from the basic funding of the university and the city. The Uniori STEAM education model has also been funded by strategic development funds of the City of Lappeenranta in the strategic period of 2018–2021.

In the Uniori model, annually, five different age groups participate in the activities (preschool, third class, fifth class, eighth class, and high school) [15]. The activities are designed taking into consideration the age and level of the participating children/young people, the content and learning objectives of the core curriculum for a certain age group, and the local resources supporting teaching and learning [15].

Implementation of the Uniori STEAM education model and its main themes of learning for each target group are described in Table 1. Studying of the main themes of the Uniori model is based on the learning objectives (O) of certain subjects in the Finnish core curriculum, and the content (C) derived from these objectives for each age group. The subjects related to the Uniori model are environmental studies and social studies at lower classes of comprehensive school and mathematics, biology, geography, physics, and chemistry at upper classes of comprehensive school.

**Table 1.** Implementation of the Uniori STEAM education model for the 3rd, 5th, and 8th grades.

| Target Group | Main Themes of Learning | Subjects, Objectives, and Contents | Transversal Competences | Pedagogy and Methodology | Strengthening Pupils' Participation and Environmental Awareness |
|---|---|---|---|---|---|
| 3rd grade | Clean water, circular economy | Environmental studies: O1, O3 and O5 + C4, C5, and C6 | T1, T3, T5 and T7 | Specialist-driven experimental learning | Being aware of one's own water footprint and basics of circular economy; consumption habits at home |
| 5th grade | Sustainable business and entrepreneurship | Environmental studies: O7 and O10 + C6 Social studies: O5 and O7 + C1 an C4 | T1, T2, T3, T5, T6 and T7 | Innovative product- and service-design learning | Strengthening entrepreneurship of pupils and production of sustainable business ideas |
| 8th grade | Clean energy and sustainable housing | Mathematics: O3 and O5+ C1 and C5 Biology: O10 and O14 + C6 Geography: O4 and O12 + C6 Physics: O4, O8, and O15 + C1, C2, C3, and C6 Chemistry: O4, O8, and O17 + C2, C3 and C3 | T1, T2, T3, T4, T5, T6 and T7 | Problem-based team learning | Being aware of one's own carbon footprint, understanding different ways of producing energy and obtaining sustainable consumption habits |

Developing pupils' transversal competences (T) is also a fundamental part of the Uniori model. These transversal skills and competences are achieved step by step as the pupils grow and as the learning units expand during basic education. Diverse pedagogical methods are used when implementing the model with pupils of different ages. The Uniori STEAM education model aims at strengthening pupils' participation and environmental awareness through activities related to topics such as water footprint, circular economy, consumption habits, sustainable business ideas, carbon footprint, and energy production [15].

In Science Education Recommendations, five principles of science education are summarized: science education is accessible and broad based, enables participation, and is collective, inspiring, and rewarding [16]. In the Uniori STEAM education model, all these principles are taken into consideration, but the value of accessibility is emphasized. In practice, it is also admitted in the recommendations that it is easiest to reach people who are already interested in science. Although inequality in education has been decreasing in several European countries, including Finland, it is still six times more likely that a person with an academic family background will participate in university education than a person without it [17]. The Uniori STEAM education model follows the recommendations from the viewpoint of accessibility: in the development of the model, a considerable amount of effort has been made to identify the needs of the target audience, but even more effort should be put into identifying the hindrances of interaction when trying to reach children and young people who are not so interested in science and academic studies [16].

## 2. Materials and Methods

This research is part of a wider research scheme that aims at evaluating and enhancing the quality and effectiveness of the Uniori educational model. The development of the research tools and measures started in spring 2019 with the preparation of a student survey to be submitted yearly to the ninth grades in the schools implementing the Uniori activities [14]. This study uses the survey data from years 2020 and 2021 to understand young people's attitudes toward and perceptions of sustainable development, and to evaluate the principles and actions of the Uniori model in terms of education for sustainability.

### 2.1. Research Questions

The general aim of the study was operationalized with two research questions:

1.  How do the ninth graders in the city of Lappeenranta perceive sustainability?
2.  How does/can the Uniori educational model support the development of sustainability perceptions among different groups of young people?

The first research question was targeted at creating an understanding of the young peoples' perceptions of the different dimensions of sustainability and their interrelations. The second research questions aimed at finding out whether there are significant differences among the perceptions of different groups of young people. This aimed to provide a specific lens for evaluating the Uniori educational model and improve it in terms of equity and inclusion.

### 2.2. Data Collection

A survey of pupils' conceptions of and attitudes toward science, technology, sustainability, and entrepreneurship was designed based on parts of the Relevance of Science Education (ROSE) questionnaire [18], the short version of the Pupils Attitude Towards Technology (PATT) survey [19], and the EntreComp framework by the European Union [20]. The first version of the student survey consisted of six content questions, with collections of 13–30 statements and two background questions. It was piloted in spring 2019. The data were factorized with principal component analysis, and the results were used to modify statements and shorten the survey. A more detailed description of the development of the survey can be found in [14].

The modified survey was used to collect data in spring 2020. The question regarding entrepreneurialism was omitted from the survey because the respondents were administered a national survey specifically on that theme. Hence, the survey included one question regarding pupils' stand on science and technology (22 statements), one question regarding pupils' stands on environment and sustainability (18 statements), one question regarding pupils' stands on business and entrepreneurship (10 statements), one question about pupils' intentions after the ninth grade, and one question about pupil's educational aspirations. The factorization was repeated and small modifications to the wordings of some statements were made, but the number of statements and the content of questions remained essentially the same. This very slightly modified survey was administered to a new group of ninth graders in spring 2021.

In both 2020 and 2021, the survey was sent to the headmasters of all five public lower-secondary schools in Lappeenranta. The headmasters were requested to distribute the survey to all the ninth graders to be answered during classes. The survey was electronic, and pupils could either use their own mobile devices or equipment from the school to complete it. Answers were submitted anonymously, and no personal information was connected with the answers at any point. All the pupils answered the survey in Finnish.

### 2.3. Data Analysis

The collected data were first analyzed separately for years 2020 and 2021 with the variables for the questions regarding science and technology, environment and sustainability, and business and entrepreneurship factorized in their own groups. As the data for both years were noticed to factorize similarly for all three questions, the responses

from both years were combined into one dataset. The final factorizations were made using principal component analysis (PCA) with orthogonal rotation. The sampling adequacy for the three solutions was evaluated with the Keiser–Meyer–Olkin measure, and the cumulative explanative power for all solutions was noted. The reliability of the factors was estimated with Cronbach's alpha.

A set of sum variables was created based on the factorizations by calculating the arithmetic average of the values of the variables contained by each of the factors. These sum variables were then used to find statistically significant differences between different respondent groups. The statistical significance was tested with the nonparametric Wilcoxon rank-sum (also known as Mann–Whitney two-sample) test as the Shapiro–Wilk test showed that the variables were not normally distributed. The effect size of the differences was calculated using Cohen's d.

Finally, the pupils' environmental perceptions were studied further by regression analysis. A linear regression model was built for each of the three sum variables related to the environment and sustainability to see which other factors were statistically significantly related to it. The models were tested for omitted variables (RESET test) and collinearity (VIF test). All the statistical procedures were performed using the statistical software Stata.

## 3. Results

The data factorized very similarly to the data from the first version of the survey in spring 2019 [14]. The pupils' perceptions of science and technology yielded five factors, perceptions of environment and sustainability three factors, and perceptions of business two factors. The Keiser–Meyer–Olkin measure of sampling adequacy for all three solutions was greater than 0.83. The factors explained 60% of the variation in science and technology statements, 57% of the variation in environment and sustainability statements, and 69% of the variation in business-related statements. Eight out of ten of the created factors had a Cronbach's alpha greater than 0.78. The factor with the weakest reliability ($\alpha = 0.459$) was the one that reflected the critical attitude toward science and technology. The pessimistic view on environmental issues also had a slightly weaker reliability ($\alpha = 0.606$) than the other factors. The results of the factorization are presented in Table 2.

**Table 2.** Outcome of the factorization of the survey data.

| Emerging Factors by Survey Section | Number of Factors | Cumulative Explanation | KMO | Cronbach's Alpha |
|---|---|---|---|---|
| Perceptions of science and technology | 5 | 59.82% | 0.8345 | |
| Trust in science and technology | | | | 0.811 |
| Faith in science and technology | | | | 0.782 |
| Talent requirements of science and technology | | | | 0.835 |
| Gender-relatedness of science and technology | | | | 0.898 |
| Criticism of science and technology | | | | 0.459 |
| Perceptions of environmental and ecological sustainability | 3 | 56.62% | 0.8895 | |
| Consciousness of environmental issues | | | | 0.857 |
| Denial of environmental problems | | | | 0.872 |
| Pessimism about the environmental situation | | | | 0.606 |
| Perceptions of sustainable business | 2 | 69.24% | 0.9024 | |
| Financial view of business and entrepreneurship | | | | 0.791 |
| Societal view of business and entrepreneurship | | | | 0.915 |

Comparing the sum variables across different respondent groups showed no statistically significant differences between respondents from the years 2020 and 2021. There were, however, clear differences between girls' and boys' responses as well as between the responses of those who intended to continue their studies in the general upper-secondary education as opposed to those who did not intend to do so. A comparison between girls and boys is presented in Table 3, and a comparison between respondents with different educational intentions in Table 4.

**Table 3.** Girls' and boys' views on science, technology, environment, and business.

| Sum Variable | All | | | Male | | | Female | | | Cohen's d | M–W (*p*) |
|---|---|---|---|---|---|---|---|---|---|---|---|
| | N | Mean | SD | N | Mean | SD | N | Mean | SD | | |
| Trust in science and technology | 377 | 3.867 | 0.611 | 152 | 3.92 | 0.72 | 207 | 3.84 | 0.53 | 0.13 | 0.0944 |
| Gender in science and technology | 376 | 2.420 | 1.245 | 151 | 3.12 | 1.24 | 207 | 1.88 | 0.96 | 1.12 | 0.0000 |
| Talent in science and technology | 377 | 2.409 | 0.884 | 152 | 2.61 | 0.99 | 207 | 2.24 | 0.76 | 0.41 | 0.0002 |
| Faith in science and technology | 377 | 2.836 | 0.731 | 152 | 3.01 | 0.84 | 207 | 2.69 | 0.63 | 0.43 | 0.0001 |
| Criticism of science and technology | 377 | 3.318 | 0.593 | 152 | 3.26 | 0.70 | 207 | 3.37 | 0.49 | −0.18 | 0.0134 |
| Environmental consciousness | 376 | 3.831 | 0.676 | 152 | 3.71 | 0.72 | 206 | 3.95 | 0.62 | −0.36 | 0.0007 |
| Environmental denialism | 376 | 2.290 | 0.861 | 152 | 2.70 | 0.93 | 206 | 1.95 | 0.64 | 0.94 | 0.0000 |
| Environmental pessimism | 373 | 2.816 | 0.754 | 151 | 2.83 | 0.91 | 205 | 2.81 | 0.63 | 0.02 | 0.7630 |
| Financial orientation to business | 369 | 3.644 | 0.715 | 151 | 3.73 | 0.75 | 201 | 3.61 | 0.68 | 0.17 | 0.1402 |
| Societal orientation to business | 369 | 4.002 | 0.769 | 151 | 3.79 | 0.77 | 201 | 4.20 | 0.72 | −0.55 | 0.0000 |

**Table 4.** Views on science, technology, environment, and business of the pupils intending to continue their studies in general upper-secondary education (GUSE) and pupils not intending to continue to GUSE.

| Sum Variable | All | | | Continuing to GUSE | | | Not Continuing to GUSE | | | Cohen's d | M–W (*p*) |
|---|---|---|---|---|---|---|---|---|---|---|---|
| | N | Mean | SD | N | Mean | SD | N | Mean | SD | | |
| Trust in science and technology | 377 | 3.867 | 0.611 | 274 | 3.96 | 0.52 | 102 | 3.62 | 0.75 | 0.52 | 0.0000 |
| Gender in science and technology | 376 | 2.420 | 1.245 | 273 | 2.24 | 1.21 | 102 | 2.92 | 1.20 | −0.56 | 0.0000 |
| Talent in science and technology | 377 | 2.409 | 0.884 | 274 | 2.30 | 0.82 | 102 | 2.70 | 0.98 | −0.44 | 0.0001 |
| Faith in science and technology | 377 | 2.836 | 0.731 | 274 | 2.84 | 0.66 | 102 | 2.84 | 0.89 | 0.00 | 0.6304 |
| Criticism to science and technology | 377 | 3.318 | 0.593 | 274 | 3.36 | 0.53 | 102 | 3.20 | 0.72 | 0.25 | 0.0215 |
| Environmental consciousness | 376 | 3.831 | 0.676 | 273 | 3.98 | 0.61 | 102 | 3.34 | 0.68 | 0.98 | 0.0000 |
| Environmental denialism | 376 | 2.290 | 0.861 | 273 | 2.08 | 0.77 | 102 | 2.86 | 0.84 | −0.96 | 0.0000 |
| Environmental pessimism | 373 | 2.816 | 0.754 | 271 | 2.76 | 0.74 | 101 | 2.98 | 0.77 | −0.29 | 0.0134 |
| Financial orientation to business | 369 | 3.644 | 0.715 | 268 | 3.71 | 0.70 | 100 | 3.47 | 0.74 | 0.32 | 0.0030 |
| Societal orientation to business | 369 | 4.002 | 0.769 | 268 | 4.13 | 0.72 | 100 | 3.67 | 0.80 | 0.60 | 0.0000 |

Although there were statistically significant differences between girls' and boys' responses with respect to most of the sum variables, most of the differences could be regarded as small ($0.2 \leq |d| < 0.5$). A 95% trust level showed no differences in girls' and boys' trust in science and technology, pessimistic environmental view, and financial/monetary view

of business. However, girls held significantly stronger societal views of business (moderate effect size, $0.5 \leq |d| < 0.8$) and boys exhibited more strongly gendered views on science and technology as well as a stronger denial of environmental problems (large effect size, $|d| \geq 0.8$).

The responses of pupils planning to continue their studies in general upper-secondary education and pupils not heading for general upper-secondary education diverged from each other even more than the responses of girls and boys. Pupils continuing their studies to general upper-secondary school are more likely to enter tertiary education than those who do not attend general secondary education but continue to vocational education or finish school after compulsory education. Again, a great difference with respect to the degree of environmental denialism was detected, but it was accompanied by a significant difference in environmental consciousness. The respondents aiming at general upper-secondary education exhibited more environmental consciousness and less environmental denialism than the others. Differences in the moderate effect size could be seen in the gender-relatedness of science and technology, trust in science and technology, and societal orientation to business. Respondents with higher educational ambitions held less gendered views and more trust in science and technology as well as a stronger societal orientation to business than the respondents with lower educational aspirations. Statistically significant differences could be observed also in other variables, but their effect size was small.

The results of the linear regression model created separately for the three factors relating to environmental and ecological sustainability are collected in Table 5. The models for environmental consciousness and environmental denialism obtained quite a strong explanatory power, with $R^2$ greater than 0.6, whereas the explanatory power of the model for environmental pessimism was slightly less than 0.3. The VIF tests show an acceptable level of multicollinearity, but the RESET test suggests that the models for environmental denialism and environmental pessimism may lack some relevant variables.

**Table 5.** Linear regression models for environmental consciousness, environmental denialism, and environmental pessimism.

| Statistically Significant Variables | Environmental Consciousness | | | | Environmental Denialism | | | | Environmental Pessimism | | | |
|---|---|---|---|---|---|---|---|---|---|---|---|---|
| | coef. | SD | t | $p > |t|$ | coef. | SD | t | $p > |t|$ | coef. | SD | t | $p > |t|$ |
| Trust in science and technology | 0.304 | 0.041 | 7.45 | 0.000 | | | | | | | | |
| Gender in science and technology | | | | | 0.174 | 0.031 | 5.55 | 0.000 | | | | |
| Talent in science and technology | | | | | 0.220 | 0.045 | 4.90 | 0.000 | 0.235 | 0.048 | 4.92 | 0.000 |
| Faith in science and technology | 0.097 | 0.036 | 2.66 | 0.008 | 0.124 | 0.051 | 2.43 | 0.016 | | | | |
| Criticism to science and technology | 0.207 | 0.043 | 4.86 | 0.000 | 0.120 | 0.061 | 1.97 | 0.049 | 0.170 | 0.069 | 2.48 | 0.014 |
| Environmental consciousness | | | | | −0.485 | 0.055 | −8.79 | 0.000 | 0.224 | 0.064 | 3.50 | 0.001 |
| Environmental denialism | −0.251 | 0.030 | −8.48 | 0.000 | | | | | 0.275 | 0.057 | 4.79 | 0.000 |
| Environmental pessimism | 0.114 | 0.032 | 3.60 | 0.000 | 0.215 | 0.044 | 4.85 | 0.000 | | | | |
| Financial orientation to business | | | | | 0.151 | 0.046 | 3.32 | 0.001 | | | | |
| Societal orientation to business | 0.340 | 0.033 | 10.44 | 0.000 | | | | | | | | |
| Gender | | | | | −0.297 | 0.070 | −4.22 | 0.000 | 0.204 | 0.078 | 2.63 | 0.009 |
| Quality measures for the models | | | | | | | | | | | | |
| $R^2$ | 0.6489 | | | | 0.6102 | | | | 0.2926 | | | |
| ovtest (*p*) | 0.3063 | | | | 0.0000 | | | | 0.0327 | | | |
| Mean VIF | 1.43 | | | | 1.55 | | | | 1.58 | | | |

The three linear regression models show that the three different environmental perceptions are connected with different perceptions of science, technology, and business. Environmental consciousness relates to the societal orientation to business and trust in

science and technology. Yet, it is also linked, to some extent, with a critical attitude toward science and technology and only quite weakly (yet significantly) with faith in science and technology. The denial of environmental problems, on the other hand, is coupled with the financial orientation to business, a gendered view of science and technology, assumptions of certain talent requirements in science and technology, faith (instead of trust) in science and technology, and some degree of criticism of science and technology. Environmental pessimism is connected with talent requirement assumptions and criticism of science and technology. Gender alone explains some of the variation for the denialism and pessimism models but plays no role in environmental consciousness.

## 4. Discussion

The results suggest that young people hold three different kinds of orientations to environmental issues and ecological sustainability. These are coupled with different perceptions of science, technology, and business in a way that also indicates three different types of perceptions of a more holistic conception of sustainability. This, in turn, indicates that in order to affect the perceptions of and attitudes toward sustainability, a more holistic and integrated educational model compared with interventions concentrating only on one dimension of sustainability could be beneficial.

The data show significant differences in perceptions related to different dimensions of sustainability among different genders and young people with different educational intentions. Acknowledging these differences and actively taking them into account in designing sustainability education is crucial for inclusive, extensive, and effective outcomes.

### 4.1. Conscious, Dismissive, and Ambiguous Perceptions of Sustainability

The conscious perception of sustainability combines consciousness and concern for environmental problems with a societal view of business. These are accompanied with the science and technology view, which includes a fair amount of trust in the capabilities of science and technology to solve many problems, but also some criticism of science and technology. Hence, there is no blind faith in science and technology to solve all the problems. This view holds no connections with beliefs about a certain gender or talent required of people engaging in science and technology.

The dismissive perception of sustainability views things quite differently. The existence of environmental problems is largely denied, and the perception of business is focused on money. The faith in science and technology is strong, and science and technology are seen as a playground for the talented male.

The third perception of sustainability seems to hold ambiguities and confusion. Environmental pessimism is positively correlated with both environmental consciousness and environmental denialism, suggesting that it can relate to either being overwhelmed by the environmental problems or distancing oneself from them. Science and technology are seen as fields that require innate talent, which can, for its part, explain the feelings of powerlessness or reluctance in taking any actions.

In terms of the ecological dimension of sustainability, the two opposite stands on environment are compliant with the findings from twenty years ago about the ninth graders' polarized attitudes toward environmental responsibility [12]. The then-discovered connection between the positive attitude and a critical view of science and technology, and the link between the negative attitude and naturocentric environmental values are both reflected also in the results of this study. However, the view of environmental pessimism and the related confusion about sustainability was not visible in the earlier study. This suggests that in addition to the traditional cognitive and attitudinal aspects, contemporary sustainability education is perhaps facing new kinds of emotional challenges.

The perceptions of sustainability appear to be quite strongly gendered, which is also in line with previous studies [12,21]. Girls have a slightly stronger environmentally conscious orientation, and a moderately stronger societal view on business than boys. Boys exhibit a considerably stronger denialism of environmental problems and view science

and technology much more as a male area of expertise than girls. There were no notable gender differences with respect to environmental pessimism. Although both girls and boys hold views that are more aligned with a conscious than a dismissive perception of sustainability, girls' inclination toward the conscious orientation is stronger than boys'. Curiously, there was no statistically significant gender difference in the mean values of the sum variable environmental pessimism, but the linear regression analysis suggests that being a girl significantly predicts environmental pessimism.

Sustainability perceptions are also linked with the educational aspirations of the respondents. Again, both groups have a stronger conscious than a dismissive stand toward sustainability, but the difference between orientations is much stronger for the group with stronger educational interests. The respondents not intending to continue their studies into general upper-secondary education perceive science and technology more strongly as a male field requiring special talent, but also have less trust in science and technology to solve problems of humanity.

### 4.2. Uniori STEAM Education Model in the Light of the Findings

In the light of the interconnectedness of the pupils' environmental, science and technology, and business perceptions, it can be argued that an educational model integrating several STEAM subjects around a certain theme has a good potential for enhancing holistic sustainability education addressing the multiple dimensions of sustainability. Further, by changing the themes and shifting the emphasis from environmental issues in grade three to sustainable business in grade five, and to science and technology in grade eight, it is also more likely that the sustainability strengthens the multidimensional view on sustainability and inhibits the encapsulation of sustainability knowledge into a specific domains or problems.

However, the differences in the sustainability-related perceptions between girls and boys as well as pupils with different educational aspirations indicate that some accessibility concerns are still involved. Although the activities are embedded in the curriculum in a way that ensures that all the pupils are involved with them, they may still have different meanings and implications for different pupil groups. Hence, more understanding is needed on the aspects that would make sustainability education through Uniori activities even more connected with the life experiences of boys and pupils with vocational rather than academic educational interests, especially. Having a stronger A in STEAM by, e.g., introducing more elements of crafts [22] or PE education [23] to Uniori learning experiences for sustainability could provide some means for this.

### 5. Limitations

This study did not aim to create a general model of the young people's perceptions of sustainability, and the results of the statistical analyses are not generalizable to other contexts. The data are confined to one city with a specific educational model created together with the local university, strategically specialized in sustainability issues. This creates unique circumstances for the development of pupils' perceptions of sustainability, which is likely to inhibit the direct transfer of results to other schools or municipalities.

Some of the quality measures regarding the factorization and the regression analysis, such as the two Cronbach's alphas below 0.7 and the RESET test indicating omitted variables for linear regression models for environmental denialism and pessimism, also indicate some room for improvement in the creation of the local models for sustainability perceptions. The local data did not evenly include representatives from the all five lower secondary schools in the city, resulting in the possibility of the school-level aspects of the better-presented schools being overemphasized.

The main aim of the study has been to evaluate and enhance the Uniori educational model. Although the study employed quantitative and statistical methods, it is suggested that the research quality question of applicability and consistency might be better understood from the viewpoint of naturalistic rather than positivistic research paradigm. Hence,

the reader is advised to evaluate the transferability of the educational model and the significance of remarks made on the differences between pupil groups to their own educational context, instead of generalizing the perceptions of pupils to a different population, and to consider the effect of the context of this study on its results, instead of expecting the research instruments to yield similar results in different contexts [24].

## 6. Conclusions

Young people in the city of Lappeenranta hold three kinds of perceptions of sustainability: conscious, dismissive, and ambiguous. In general, adolescents are more conscious than dismissive of sustainability, and both perceptions are, to some degree, accompanied by some ambiguous or confused views on the topic. Sustainability perceptions combine various dimensions of sustainability and connect also with young people's perceptions of science and technology. The Uniori STEAM education model aims at the integration of several subjects and a wide accessibility of audiences through integration to the core curriculum. The Uniori activities reach out to all the pupils in the city on several occasions during their basic education. Yet, in terms of effectiveness, it seems that equal access to activities does not guarantee equal access to learning about sustainability. Hence, in developing the Uniori STEAM education model, more emphasis must be put on enhancing the ecosocial ability of diverse pupil groups.

**Author Contributions:** Conceptualization, J.N. and L.J.; methodology, J.N.; software, J.N.; formal analysis, J.N. and L.J.; writing—original draft preparation, J.N. and L.J.; writing—review and editing, J.N. and L.J.; visualization, J.N.; All authors have read and agreed to the published version of the manuscript.

**Funding:** The creation of the survey instrument was funded by the city of Lappeenranta as part of the cooperation No official research funding has been received for this study.

**Institutional Review Board Statement:** Research followed the LUT guidelines for good scientific practice and no personal information was collected. Hence according to the LUT code of conduct an ethical approval was not required.

**Informed Consent Statement:** School consent was obtained with a contract between LUT university and city of Lappeenranta.

**Acknowledgments:** The authors wish to thank the city and the lower secondary schools of Lappeenranta for enabling the data collection.

**Conflicts of Interest:** The authors declare no conflict of interest.

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
