# Peer review of "Toward Integrated and Inclusive Education for Sustainability with School–University Cooperation"

_sustainability, doi:10.3390/su132212486_

Round 1

Reviewer 1 Report

The analysis carried out, as well as the method followed are optimal, and conform to the standard methodological parameters.
The issuance of conclusions is valid, but it suffers from the same problem as the entire article: the low representativeness and the localism of the study. A study such a local level cannot be extrapolated, nor can they be deduced from the same general paradigms.
The great problem of the article is precisely that, reducing itself to a study in a single city, with a great influence from the LUT University itself (a pioneer in the sector in Finland), and which substantially modifies the perception of the population living in it. environment, and even more so from secondary schools. It is not considered publishable precisely because of this aspect, not because of its quality in the research, but because of the smallness of the research environment and its low representativeness.

Author Response

Thank you for this valuable feedback. We understand your concern about the generalizability of the results due to the locality and peculiarity of the context of our study. We have attempted to address this issue by reformulating the research question 1 and adding a Limitations -section to the manuscript. We agree, that the results are not generalizable as such, but we also argue that it has not been the aim of the research either. The main aim of the research has been to present and evaluate the Uniori educational model. Hence the results, which are of interest to the scientific audience relate to the educational model rather than on the pupils’ perceptions of sustainability as such. We also suggest that the quality of the research should be evaluated more from the naturalistic than positivistic research paradigm.

We have also attempted to improve the research design and the discussion of findings by reformulating the abstract, revisiting the research questions and adding the limitations -section. The changes to the manuscript have been marked with red font.

Reviewer 2 Report

 This study presents a new STEAM education model developed to support the development of ecosocial attitudes and abilities. Through empirical analysis of pupils’ perceptions of sustainability, study evaluates the educational model from the viewpoint of sustainability education and suggests some ideas for future development of the model. The paper has an empirical part which consists of using three instruments and applied some statistical tests to answer its research questions.

The paper in interesting and it has the potential to contribute to the current literature, however has some drawbacks which need to be faced. The presentation is hard to follow and I think the paper has to be reorganized some way to be more comprehensible to the reader. I suggest revision because there some part to be improved.

 -First the abstract is very poor and underdeveloped. It needs to be rewritten focusing and including what the research part and the finding.

-Some separate section reporting clearly what are the research questions or hypotheses would be useful. Pay attention to avoid vague expressions such as “..What kind of attitudes ..”. It needs to be more specific.

-In the paragraph with the validation issues, please report the method used (e.g. PCA or other).

 -correct R2 to R2   

-Table 5, where results from linear regression are reported, needs to include the statistical significance of the coefficients (SD, t , p-values)

- The limitations of the study should be reported.

Author Response

We wish to thank for the valuable and precise comments. We have attempted to meet all of the requests/suggestions and hope that the manuscript now meets the expectations of the reviewer.

Here are the changes made based on the comments:

  • The abstract has been rewritten with a better focus on the research process and results.
  • A separate section for research questions was added and the first research question was reformulated. The motives behind the two research questions were more clearly explicated.
  • It was already mentioned in the text that the factorization was conducted with a principal component analysis, but this was further emphasized by adding the abbreviations PCA to the text. It hopefully makes it also more visible.
  • R2 was corrected to R2.
  • Columns and values for SD, t, and p-values were added to the Table 5 for all of the three regression analysis.
  • Limitations -section was added and limitations of the study more clearly addressed.

The changes made to the manuscript have been marked with red font. We hope that the changes made to the abstract and the added sections for research questions and limitations have made the presentation easier to follow and comprehend.

Round 2

Reviewer 1 Report

The changes made add a new dimension to the article and clarify its usefulness. It is true that its contribution is limited, depending on the scope of the research and its sample, but its model, as a case study, may be of interest to the scientific community dedicated to learning systems.

Reviewer 2 Report

The revised paper is improved and addresses all comments

I am positive for its publication